# Integration of Omics Data and Network Models to Unveil Negative Aspects of SARS-CoV-2, from Pathogenic Mechanisms to Drug Repurposing

**DOI:** 10.3390/biology12091196

**Published:** 2023-08-31

**Authors:** Letizia Bernardo, Andrea Lomagno, Pietro Luigi Mauri, Dario Di Silvestre

**Affiliations:** Institute for Biomedical Technologies—National Research Council (ITB-CNR), 20054 Segrate, Italy; letizia.bernardo@itb.cnr.it (L.B.); andrea.lomagno@istitutotumori.mi.it (A.L.); pierluigi.mauri@itb.cnr.it (P.L.M.)

**Keywords:** omics, networks, drug repurposing, SARS-CoV-2, COVID-19

## Abstract

**Simple Summary:**

SARS-CoV-2 caused the COVID-19 health emergency, affecting millions of people worldwide. Samples collected from hospitalized or dead patients from the early stages of pandemic have been analyzed over time, and to date they still represent an invaluable source of information to shed light on the molecular mechanisms underlying the organ/tissue damage. In combination with clinical data, omics profiles and network models play a key role providing a holistic view of the pathways, processes and functions most affected by viral infection. In fact, networks are being increasingly adopted for the integration of multiomics data, and recently their use has expanded to the identification of drug targets or the repositioning of existing drugs.

**Abstract:**

Severe acute respiratory syndrome coronavirus 2 (SARS-CoV-2) caused the COVID-19 health emergency, affecting and killing millions of people worldwide. Following SARS-CoV-2 infection, COVID-19 patients show a spectrum of symptoms ranging from asymptomatic to very severe manifestations. In particular, bronchial and pulmonary cells, involved at the initial stage, trigger a hyper-inflammation phase, damaging a wide range of organs, including the heart, brain, liver, intestine and kidney. Due to the urgent need for solutions to limit the virus’ spread, most efforts were initially devoted to mapping outbreak trajectories and variant emergence, as well as to the rapid search for effective therapeutic strategies. Samples collected from hospitalized or dead COVID-19 patients from the early stages of pandemic have been analyzed over time, and to date they still represent an invaluable source of information to shed light on the molecular mechanisms underlying the organ/tissue damage, the knowledge of which could offer new opportunities for diagnostics and therapeutic designs. For these purposes, in combination with clinical data, omics profiles and network models play a key role providing a holistic view of the pathways, processes and functions most affected by viral infection. In fact, in addition to epidemiological purposes, networks are being increasingly adopted for the integration of multiomics data, and recently their use has expanded to the identification of drug targets or the repositioning of existing drugs. These topics will be covered here by exploring the landscape of SARS-CoV-2 survey-based studies using systems biology approaches derived from omics data, paying particular attention to those that have considered samples of human origin.

## 1. Introduction

Severe acute respiratory syndrome coronavirus 2 (SARS-CoV-2) is an enveloped positive single-stranded RNA virus belonging to the Coronavirus family [1,2]. It triggered the COVID-19 health emergency, which, to date, has affected more than seven hundred million people worldwide and caused approximately seven million deaths (Figure 1A).

To infect and spread, SARS-CoV-2 employs its surface spike (S) protein, which interacts with the host’s angiotensin-converting enzyme 2 (ACE2) receptor [3,4]. ACE2 is present on the surfaces of different cell types, and the viral entry mediated by the S–ACE2 interaction was the first evidence of the mechanism underlying the infection in human beings [5]. Due to the correlation between ACE2 expression and COVID-19 patient outcomes, the analysis of ACE2 across different tissues/organs has been useful in deciphering the potential routes of SARS-CoV-2 infection and damage [4]. Other studies have investigated its variation in several pathological conditions, revealing that COVID-19 patients with common comorbidities of cancer and chronic diseases may show higher ACE2 expression levels, which could lead to an increased susceptibility to multi-organ damage [6]. On the other hand, in cells with lower ACE2 expression, it has been reported that Neuropilin-1 (NRP-1) acts to facilitate virus entry [7,8], while a second route involves the endolysosomal pathway via membrane fusion at the cell surface through the cathepsin L1 protein [9,10,11].

Following SARS-CoV-2 infection, a spectrum of disease ranging from asymptomatic to the manifestations of highly severe symptoms has been observed. Bronchial and pulmonary cells, involved at the initial stage, trigger a hyper-inflammation phase, damaging a wide range of human organs and tissues, including the heart, blood vessels, brain, liver, intestine and kidney [11,12,13]. Under the condition of the serious threat to public health and the global economy, many efforts were initially dedicated to the sequencing and monitoring of the SARS-CoV-2 variants that gradually emerged from the virus’ genomic evolution (Figure 1B). Combined with mobility data, this information has been useful in mapping the outbreak trajectory [14]. At the same time, many efforts have been devoted to the rapid search for effective therapeutic strategies, involving hundreds of potential drugs and thousands of patients in clinical trials [15].

Since the early outbreak phases, epidemiological data and clinical specimens have been collected from hospitalized and/or dead patients. Today, they still represent an invaluable resource that the scientific community is analyzing and evaluating to shed light on the molecular mechanisms underlying organ/tissue damage and to offer opportunities for diagnostics and therapeutic designs. To achieve this goal, the contribution of omics technologies is essential. In fact, genomics, transcriptomics, proteomics and metabolomics have now reached a level of effectiveness and depth that allows for a holistic evaluation [16]. However, multiomics data integration and their interpretation remain a challenge. To address this, network-based approaches provide a framework to represent and process the interactions between omics layers in a graph, which simulates the molecular wiring in a cell [17]. In the context of SARS-CoV-2 investigation, a milestone in these strategies’ development was the release of the first network model, which shed light on host–virus protein interactions [18]. Due to the urgent need for clinical and pharmacological solutions, as well as the lack of animal models to mimic the disease’s pathogenesis and its treatment [19], this and other protein–protein interaction (PPI) network models have been used for drug repurposing purposes [20,21,22]. In particular, the combination of these models and experimental omics data has often been considered in the field of network science to identify relevant molecules, such as hubs, as candidate targets for pharmacological treatments [23,24,25,26].

Based on these premises, we aim to explore here the landscape of studies based on the combination of omics data and network analysis to investigate the molecular mechanisms underlying SARS-CoV-2 infection and triggering organ and tissue damage. Special attention is paid to studies where the authors have characterized the omics profiles of real human samples from control and patient cohorts, as well as to those that have exploited the network topology for drug repurposing or for the characterization of new drug targets. Thus, the methods and strategies adopted are dissected into their most relevant steps and aspects.

## 2. Word Associations in Titles of Manuscripts Focused on SARS-CoV-2 Investigation

The semantic analysis of the manuscripts published in the last three years has provided a good snapshot of how the scientific community has modulated its activity in the various phases that have characterized the pandemic. Some terms, including “detection”, “spike protein (S)”, “hospital/hospitalization” and “ACE2”, were always displayed. In particular, the association that emerged in the first semester between the terms “detection” and “test” suggested the need to develop rapid methods to identify the presence of the virus (Figure 2A). In addition to “hospital/hospitalization”, the pressure suffered in the hospital context was also highlighted by the term “healthcare worker”, which showed a high frequency until the first semester of 2022 (Figure 2A–D). Equally interesting are the terms that provide evidence of the development and geographical spread of the pandemic. Countries like Italy and China are present in the titles of manuscripts published in the initial phase. Meanwhile, other countries, such as Japan, Brasil, Spain, Germany, India, France and England, appear in the following semesters. In both semesters of 2021, we observe that the incidence of terms such as “variant”, “mutation”, “transmission” and “diffusion” begins to grow, while the terms dominating 2022 evidence the different variants that evolved and the attention paid to the immune system, related also to vaccine and drug development (Figure 2B–E). It should be noted that the term “vaccine” immediately appeared as a potential solution, and almost immediately it was specifically associated with mRNA. On the contrary, the generic term “drug” was present in the first four semesters and disappeared in the last two, when vaccines were developed and administered (Figure 2E,F). Of note, the role of omics and network analyses, the topics of our review, did not significantly emerge from the titles of the manuscripts that we considered. However, in the first and last semester, we noticed terms related to in silico approaches, which, as we will see later in our review, could be associated with drug repurposing studies through network analysis. In the same way, they could be also associated with the “molecular docking” term, which was present in several semesters.

## 3. Human Samples and Remodeling of Omics Profiles Following SARS-CoV-2 Infection

A key point in evaluating the global impact of SARS-CoV-2 on tissue and organ physiology is the multiomics profiling of samples collected by COVID-19 patients [4,6,11]. In the last 3 years, and particularly in 2021 and 2022, a number of studies based on omics data have been published, helping the scientific community to gain a deeper understanding of SARS-CoV-2 infection, as well as of the molecular mechanisms that it activates in COVID-19 patients (Figure 3). Thanks to the improved sequencing capabilities, public resources and databases are mainly enriched in genomics, transcriptomics and proteomics data related to SARS-CoV-2. After their generation, the major challenge remains their efficient and quick integration and interpretation [27]. Of note, their retrospective evaluations, or new analyses of previously collected samples, represent a valuable source of information to be combined with molecular network models. They allow for the analysis of the omics complexity in the simplest way, and for the processing of the characterized profiles by taking into account the relationships among molecules [28].

Despite the importance of having a complete picture of the processes influenced by viral infection, to date, few studies have applied truly multiomics approaches, and even fewer on tissues or organs. Some of them have combined proteomics and metabolomics in plasma [29,30] and urine [31], or proteomics and lipidomics [30,32] in plasma. In this scenario, Druzak and colleagues investigated the molecular mechanisms that occurred in adult and pediatric subjects affected by COVID-19 and in children with multi-inflammatory syndrome (MIS-C) [30]. Their findings showed differences in key mediators of pathogenesis, highlighting the role of fibrinogen in red blood cell aggregation and a strong correlation of cytokine upregulation with disease severity in pediatric populations. Meanwhile, pro-inflammatory pathways and complement and coagulation cascades were upregulated in both children and adults. Complement and coagulation cascades, platelet aggregation, myeloid leukocyte activation and arginine and proline metabolism were observed by Survana et al. during progression from a non-severe to a severe state [29]. They showed that the transforming growth factor beta-1 proprotein (TGFB1) and the galectin-3-binding protein (LGALS3BP) were significantly downregulated in severe patients, correlating this result with the progression of fatal COVID-19 infection. Meanwhile, they hypothesized increases in creatine and arginine as indicators of kidney dysfunction and heart failure. Human plasma was the collected biofluid also in Lam et al.’s work, who relied on lipidomics and proteomics [32]. They focused their attention on monitoring the compositional variation in exosome-enriched extracellular vesicles (EVs) from patients at different temporal stages of COVID-19. EVs’ lipid membrane fluidity was affected by the dysregulation of the raft lipid metabolism. This impacted the localization of the protein cargo, which in turn had distinct biological attributes based on the temporal clinical stage. In agreement with Suvarna and Druzak’s studies, the complement and coagulation cascades and platelet activation were the most dysregulated pathways, confirming their role in the innate response against viruses. Moreover, the enrichment of presenilin-1 (PS-1) in the EVs from the hyper-inflammatory phase indicated this protein as a marker of distinct cellular responses in recipient cells.

Due to the spread of SARS-CoV-2 infection, many efforts have also been dedicated to the investigation of maternal, fetal/placental and neonatal immunity, even following vaccination [33,34,35]. Comparative studies have highlighted alterations in the activation of systemic cytokines and peripheral leukocytes between pregnant and non-pregnant COVID-19 patients [36,37]. The understanding of the pregnancy-specific response to SARS-CoV-2 was enriched by a high-throughput proteomics-based study of human plasma [38]. It was collected from 101 pregnant women (72 diagnosed with COVID-19 and 29 who tested negative) and 93 non-pregnant individuals (53 diagnosed with COVID-19 and 41 who tested negative). Over 7000 proteins were identified, and 708 were differentially abundant following Uniform Manifold Approximation and Projection (UMAP) analysis. The extracellular matrix, immune response and viral infection were the most enriched processes. Terms related to protein transport, translation, platelet activation, vascular endothelial growth factor A (VEGF) and platelet-derived growth factor subunit A (PDGF) were enriched in non-pregnant cases. Although some processes, including cell adhesion, wounding and blood coagulation, were shared by both groups, the results overall suggested a stronger response to infection in non-pregnant women, whereas pregnant cases showed a modulated response directed toward the protection of the fetus from inflammation.

Biofluids and omics technologies have been combined also for prognostic purposes. Chen et al. defined the transcriptome, metabolome and proteome in blood samples from COVID-19 patients [39]. Their goal was the discovery of biomarkers for the prognosis of SARS-CoV-2 infection provoking multi-organ damage. By analyzing three different infection time points, they associated differences in T-cell mobilization with the control pro-inflammatory response. Matrix metalloproteinase-9 (MMP9), C-X-C motif chemokine 2 (CXCL2) and C-X-C motif chemokine 6 (CXCL6) were proposed as upregulated markers of fatal progression. Moreover, different biomarkers per omics dataset were extracted, including the let-7 family from exRNA, S-acyl fatty acid synthase thioesterase, the medium-chain (OLAH) and T-cell surface glycoprotein CD3 epsilon chain (CD3E) from mRNA and the complement C4-A (C4A) and complement C4-B (C4B) proteins, as involved in T-cell activation and the suppression of inflammation. Biomarkers belonging to similar protein families were reported by Bi et al., who proposed to use urine samples and a machine learning approach to follow disease progression [31]. Among the discovered markers, C-X-C motif chemokine 14 (CXCL14), transforming protein RhoA (RHOA), ras-related C3 botulinum toxin substrate 1 (RAC1) and cubilin (CUBN) were validated by parallel reaction monitoring (PRM); interestingly, CXCL14 was downregulated in the urine of severely ill patients and correlated with blood lymphocytes.

In addition to biofluids, some contributions have explored the effect of viral infection using specific cell types [40,41,42,43]. Xu et al. analyzed Calu3 cells at 24 h post-infection, characterizing their transcriptome, proteome and ubiquitinome profiles [40]. More than two thousand genes and one thousand proteins were differentially regulated. In this context, the role of pro-inflammatory processes emerged through the modulation of pathways like p38/MAPK, PI3K-AKT, EGFR/VEGFR, TLR and TNF. Furthermore, the results suggested the relevance of ubiquitination for viral infection. Indeed, multiple ubiquitination sites have been identified on proteins participating in the TLR and TNF signaling pathway, while three sites have been found in the spike (S) protein. Calu3 cells were also analyzed by Pinto et al. at the transcriptomic, proteomic, acetylomic, phosphoproteomic and exometabolome levels [43]. By combining this multitude of analyses, they demonstrated that the induction of the type I IFN response, the activation of the DNA damage response and Hippo signaling dysregulation were SARS-CoV-2 Norway/Trondheim-S15 strain infection time-dependent. Of note, the results showed also an interplay among phosphorylation and acetylation dynamics in host proteins, and its effect on the altered release of metabolites, especially organic acids and ketone bodies. The proteome profiles of the Calu3 and Caco2 cell lines at different time points were used by Caccuri et al. to demonstrate the competition between virus quasi-species (MB61^0^ and MB61^222^ patient isolates) to maintain dominant replicative activity and spread by manipulating the host cell’s innate immunity. The same group investigated the capability of SARS-CoV-2 to infect human primary lung microvascular endothelial cells (HL-mECs), supporting the hypothesis of a direct role of SARS-CoV-2-infected HL-mECs in sustaining vascular dysfunction during the early phases of infection. In support of this, they observed the release of pro-inflammatory and pro-angiogenic molecules, as well as the expression of antiviral molecules such as annexin A6 (ANXA6) and Interferon-induced GTP-binding protein Mx1 (MX1) [41].

The results obtained by analyzing these cell lines can be related to human tissues and biofluids; thus, they can be used to evaluate both disease states and drug efficacy. Of course, it is important to underline the limitation of analyzing a single cell line, decontextualized from its natural environment made up of molecular and cellular interactions. The analysis of biofluids, mainly serum and plasma, gives results that reflect the states of different sources, including cell lines and tissues. Thus, they provide a more systemic view. For this purpose, specific tissues certainly represent well-defined, complex systems whose multiomics profiles can be better associated with a disease state, as well as with therapeutic efficacy. However, in comparison to biofluids or infected cell lines, omics studies on tissues and organs from patients affected by COVID-19 appear much less represented in the literature.

The proteome, phosphoproteome and transcriptome were analyzed by Cantwell et al. in the heart, kidney and lung in a Syrian hamster model [19]. Meanwhile, other studies have taken into consideration large cohorts of human samples but focusing on ACE2 expression in different organs/tissues and pathological conditions [4,6]. As for human tissues and proteomic landscapes, a major study analyzed 144 autopsies from seven different organs, such as the lung, spleen, liver, kidney, heart, testis and thyroid [11]. Taking non-COVID-19 cases as a reference, 5336 were found to be dysregulated. For each analyzed organ, hierarchical clustering analysis revealed a clear separation between infected and non-infected samples. However, excluding the testis [44], they shared only 27 dysregulated proteins, suggesting an organ-specific response. This set included the CRP and CD163 proteins, associated with the hyper-inflammatory response and tissue repair. Both the spleen and lung exhibited a similar immune response pattern where the downregulation of the tyrosine-protein kinase Lck (LCK) could indicate the suppression of T-cell-mediated responses. Two immune checkpoints, carcinoembryonic antigen-related cell adhesion molecule 1 (CEACAM1) and CD276 antigen (CD276), were upregulated in the lung, suggesting the suppression of adaptive immunity. Meanwhile, in the spleen, it was suggested by the dysregulation of proteins enriched in the PD-1 and PD-L1 pathway, along with the inhibition of B-cell receptor signaling. Among the considered COVID-19 cases, 14 out of 19 were characterized by systemic hyper-inflammatory and multiple organ dysfunction syndrome (MODS). Their livers showed the enrichment of processes involved in the acute phase response, cytokine secretion and neutrophil degranulation. To gain a better understanding of the mechanisms exerted by SARS-CoV-2, the authors selected six functional protein clusters, including viral receptors and proteases, transcription factors (TFs), cytokines and their receptors, the complement and coagulation system, angiogenesis and fibrosis markers, as reported in other studies [29,30,42,45,46]. TFs were strongly correlated with the activation of the inflammatory response, and 395 out of 1117 were found altered; they were enriched also in processes involved in the spliceosome, viral carcinogenesis, tissue injury and hypoxia. Similarly, 112 dysregulated cytokines were enriched in pathways including angiogenesis and the growth factor response. Finally, in addition to ACE2 [3], Nie and colleagues discovered several potential molecules involved in coronavirus entry. They included C-type lectin domain family 4 member M (CLEC4M) [47], CD209 antigen (CD209) [48], NPC intracellular cholesterol transporter 1 (NPC1) [49], CEACAM1 [50] and procathepsin L (CTSL) [51]. ACE2, CD209 and CLEC4M did not show significant dysregulation in the lung, while ACE2 was downregulated in the kidney and heart, potentially impacting its modulatory roles in angiotensin II and related processes, such as inflammation, vasoconstriction and thrombosis [52]. On the contrary, CTSL, a serine protease involved in the endosomal pathway of SARS-CoV-2, was significantly upregulated in the lung, spleen, kidney and thyroid, suggesting it as a potential therapeutic target [51].

## 4. Omics Data-Derived Molecular Network Strategies to Explore SARS-CoV-2-Induced Organ/Tissue Damage, Identify Drug Targets and Reposition Existing Drugs

The essential basis of systems biology is considering a biological phenomenon as an ensemble of elements dynamically interacting at different levels. Based on this assumption, the complex network of interactions between DNA, transcripts, proteins and metabolites is considered the decisive cause of the emergent properties that determine cellular and/or tissue/organ dysfunction [53]. In the fight against the pandemic, a relevant contribution to this field was provided by Gordon et al., who released the first viral–human protein–protein interaction (PPI) network for SARS-CoV-2 [18]. By cloning, tagging and expressing 26 out of 29 SARS-CoV-2 proteins in human cells, the authors identified 332 high-confidence PPIs between SARS-CoV-2 and humans. Among them, 66 proteins were druggable by 69 compounds, 29 of which were already approved by the US Food and Drug Administration [54]. For a similar purpose, Zhou et al. used high-throughput yeast two-hybrid experiments and mass spectrometry [55]. Following their pipeline, they built a comprehensive SARS-CoV-2–human PPI network consisting of 739 high-confidence interactions, and almost half of them were new. The interaction partners found showed a large overlap with already published datasets and DEGs in samples from COVID-19 patients. In addition, by exploiting the network proximity measure [56], more than 2900 drugs were in silico tested and 23 showed significant outcomes. Of note, carvedilol was effective in treating COVID-19 patients due to its antiviral properties, demonstrated in a human lung cell line infected with SARS-CoV-2. Using two large independent COVID-19 patient databases, they found that its use was associated with a lowered risk (17–20%) of a positive COVID-19 test, validating the trustworthiness of the proximity measure.

Two additional studies helped to unravel the range of interactions between the virus and host. One of them, published by Das et al., was based on codon usage patterns between pairs of co-evolved host and viral proteins [57]. In addition to inferring both negatively and positively interacting edges, the authors found the multi-domain non-structural protein 3 (NSP3) and spike (S) as the most influential proteins in interacting with multiple host proteins. The MAPK pathway was the most affected during SARS-CoV-2 infection, while proteins participating in multiple pathways were, as expected, central in host PPIs and mostly targeted by multiple viral proteins. On the other hand, Schmidt and colleagues dedicated their efforts to characterizing the interactions between SARS-CoV-2 RNAs and the proteins of host cells [58]. Following RNA antisense purification and mass spectrometry, the authors identified 104 human proteins interacting with and affecting the virus RNAs. The correlation with proteome modulation allowed the characterization of pathways with a role in SARS-CoV-2 infection, as well as the identification of two viral RNA binders, cellular nucleic acid-binding protein (CNBP) and La-related protein 1 (LARP1), able to restrict SARS-CoV-2 replication in infected cells. In this context, the pharmacological inhibition of some RNA interactors, including peptidyl-prolyl cis-trans isomerase A (PPIA), sodium/potassium-transporting ATPase subunit alpha-1 (ATP1A1) and the ARP2/3 complex, showed effectiveness in decreasing viral replication in cell lines.

PPI and co-expression network models have been combined with experimentally determined omics profiles to organize them into functional, topological and disease modules [28,42]. The structure of these networks was analyzed to select hub nodes as key molecules underlying the pathophysiological processes triggered by viral infection and thus as potential drug targets [24,25,26]. This landscape is dominated by studies that have adopted common pipelines characterized by the combination of different complementary approaches and tools, including databases (Table 1), omics data, network models, algorithms for network topology, molecular docking and molecular dynamics (MD) simulation (Figure 4). Several of these contributions fall within the network pharmacology area, which has attracted the attention of modern science, gaining even more interest in light of the urgent need for effective therapeutic strategies that target SARS-CoV-2 and/or human proteins to control the viral infection [15].

The pandemic has led to a growing trend in the use of herbal drugs, with therapeutic and antiviral properties and low side effects. After the use of Traditional Medicine (TM), botanical formulations and cellular systems have been recently analyzed in a retrospective manner with the aim of identifying hidden aspects of SARS-CoV-2’s pathogenic mechanisms, as well as to characterize active compounds and drug target candidates. The latter have been selected among hubs found by processing topologically the PPI network models reconstructed from experimental omics profiles (Figure 4). However, most of them have used differentially expressed genes (DEGs) [23,25,26,74,75,76,77,78,79,80,81,82,83,84,85], while few have relied on proteomics [86] or metabolomics data [87]. Interestingly, similar strategies have been used for network pharmacology studies to evaluate comorbidities. As a matter of fact, SARS-CoV-2 infection has been investigated in correlation with mucormycosis [88], influenza [24], diabetic kidney disease [83], multiple myeloma [84], cardiomyopathy [77,89,90], non-alcoholic fatty liver disease (NAFLD) [79], pulmonary fibrosis [91], colon cancer [92], lung cancer [82,85], glioblastoma [93] and HIV [94].

From the studies dedicated to the treatment of SARS-CoV-2 by TM, the relevance of some active compounds and drug targets has emerged more frequently than others. In particular, quercitin [88,90,92,95,96] and kaempferol [88,91,95,96], belonging to the flavonoid family, were the active phyto-compounds most mentioned as effective. Along with them, wortmannin [23], a fungal metabolite, amygdalin [97], a cyanogenic glycoside found in many plant seeds, phillyrin [24], an endophytic fungal isolate, cepharanthine [26], an anti-inflammatory and antineoplastic compound isolated from the Stephania plant, luteolin [96], a flavonoid with antioxidant properties and a free radical scavenger, β-sitosterol, estrone, and stigmasterole from ephedra bitter almond [98] and phaseolinisoflavan, glabrene, shinpterocarpin and irisolidone [95] were investigated and considered among the main active compounds. In the same way, a role as a hub and drug target appeared recursive for some specific genes. In addition to ACE2, Interleukin-6 (IL6) [79,90,96,97,99,100], vascular endothelial growth factor A (VEGFA) [23,77,97,100,101], transcription factor p65 (RELA) [75,90,91,94,96,101], signal transducer and activator of transcription 1-alpha/beta (STAT1) [26,75,83,93,100], mitogen-activated protein kinase 1 (MAPK1) [16,91,94,96,100], MAPK8 [90,91,96,100], mitogen-activated protein kinase 3 (MAPK3) [91,94,97], proto-oncogene tyrosine-protein kinase Src (SRC) [26,91,97], epidermal growth factor receptor (EGFR) [90,91,97] and RAC-alpha serine/threonine-protein kinase (AKT1) [78,91,94] were among those best ranked following network topology, molecular docking and molecular dynamic analyses—all results that highlight the enrichment and involvement of the MAPK [23,100,101], PI3K-Akt [24,26,97,99] and Interferon signaling pathways [41,42,75,76,83].

## 5. Network Topology: From Hubs to Proximity Measure by Way of Shortest Paths

Due to the long lead times for vaccine development, where the typical timeline for approval can exceed 10 years, the emergence of the COVID-19 pandemic has triggered several studies on the reuse of known drugs to accelerate the adoption of new therapies against SARS-CoV-2 infection. One of the first studies, published in 2020, profiled a library of 12,000 drugs, including clinical-stage and Food and Drug Administration (FDA)-approved molecules [102]. One hundred molecules inhibited viral replication and 21 were dose-responsive. Most of the molecules identified in this study were at an advanced clinic stage, ensuring a pharmacological safety profile, which was important to accelerate the clinical evaluation of these drugs for the treatment of COVID-19.

As reported in the previous paragraphs, the authors of several manuscripts have concentrated their efforts on the investigation of the molecular network structure in terms of characterizing nodes, called hubs, as candidate targets for drug therapies. A hub node represents the signature of a deeper organizing principle called a scale-free property [28]. These strategies arise from the assumption that the underlying architecture of a network enables the cellular functions to be carried out by a system (cell, tissue, organ) under a given condition [53]. To provide a simplified picture of such concepts, a molecule (i.e., gene, transcript, protein, metabolite) is relevant, or central, if it is close to and connected with many other molecules. To evaluate these features and identify the molecules most topologically relevant in complex networks including thousands of nodes and interactions, different parameters, called centralities, are available; a comprehensive description may be found in Vella et al. [28] or Scardoni et al. [103]. In addition to the degree, which is the simplest centrality, one of the most used is known as betweenness (Figure 5). Nodes with a high value of this centrality maintain communication with groups of nodes and control the information flow that passes through them. In other words, this hub or bottleneck represents a fundamental element of connection in signaling pathways, thus becoming a candidate drug target, as reported in several studies focused on SARS-CoV-2 investigation [23,24,25,26,74,75,76,77,78,79,80,81,82,83,84,85,86,89,90,91,92,93,94].

Betweenness centrality, like other topological parameters, is based on the shortest path measure, i.e., the shortest path that connects two nodes in a network model. This measure underlies also the drug–disease proximity measure that quantifies the interplay between drug targets and diseases and supports a mathematical approach employed to infer potential new drug candidates [56]. In addition to the network proximity measure, previously published algorithms used for drug repurposing, like DrugNet [104] and GPSnet [105], or specifically designed for COVID-19, like SAveRUNNER [106], CovidX [107] and LUNAR [108], have proven to be powerful in examining lists of existing and approved drugs.

In Guney et al., the proximity measure was applied to test 238 drugs in 78 diseases for a total of 402 combinations. Their idea was born from the question of whether drugs that target proteins/genes (drug targets) closer to the disease proteins/genes are more effective than drugs that target distant proteins/genes. Moreover, they assumed that genes associated with a disease tend to cluster in the same network neighborhood, called the disease module [109,110], as well as that the impact of drugs is typically local and thus restricted to proteins within two steps in the interactome. Based on these assumptions, the authors’ results suggested that 15% of the drug–disease associations among the drug targets corresponded to disease proteins, while, for 59%, the drug targets were proximal to the disease proteins/genes. Proximal drugs tended to involve the endocrine system and metabolic processes, whereas distant drugs were enriched in the anti-inflammatory and pain relief-related categories. Moreover, the proximity distance was found to be a good measure of a drug’s efficacy, with proximal drugs more likely to be therapeutically beneficial than distant drugs, which usually correspond to palliative treatments [56]. The same authors further used and validated this approach in investigating the associations between over 984 FDA-approved drugs and 23 types of cardiovascular diseases (CV). By integrating network proximity and large-scale patient-level longitudinal data complemented by mechanistic in vitro studies, already known relationships between drugs and drug targets, like side effects, were found. Relying on the FDA-approved CV drugs (177 out of 984) and their known CV indications, the approach provided a value of accuracy over 70% (AUC), outperforming other network distance-based measures between drug targets and disease modules [111].

The availability of the proximity measure during the COVID-19 pandemic was important in prioritizing approved drugs to treat SARS-CoV-2 infection without regard for their established disease indications [20]. In their work, Morselli Gysi and colleagues implemented three different network-repurposing strategies relying on artificial intelligence (Al), network diffusion and network proximity. Based on these algorithms, 12 pipelines were used to test 6340 drugs and rank their potential efficacy against SARS-CoV-2. To obtain consistent, reliable outcomes across all datasets and metrics, the authors selected 918 drugs for which all pipelines offered predictions and whose compounds were available in their library to be experimentally tested. In fact, the predictions obtained were experimentally screened in VeroE6 cells derived from the African green monkey. Specifically, 35 drugs were cytotoxic, 37 had a strong effect and 40 had a weak effect, while the rest did not show detectable effects on viral infectivity. Subsequently, the authors limited the analysis of the 918 screened drugs by considering as positive 37 drugs in clinical trials and as negative the remaining 881. Unfortunately, since the outcomes of these trials were largely unknown, each pipeline was tested referring to the pharmacological consensus of the medical community. AI-based pipelines showed the best predictive power for the drugs selected for clinical trials, while proximity offered better predictions for the experimental outcomes of the screened 918 drugs. Since these predictive models were limited by finite experimental resources, the authors combined the predictions of the different pipelines by implementing three heuristic rank aggregation algorithms. Among them, CRank [112], which relies on Bayesian factors, showed the highest predictive performance for all datasets, exceeding the predictive power of the individual pipelines. Among the drugs ranked by CRank, those that had a positive effect on VeroE6 cells were further tested in Huh7 cells. In addition to chloroquine and hydroxychloroquine, which have been tested repeatedly in the literature, auranofin, azelastine and digoxin showed very strong anti–SARS-CoV-2 responses. However, they were not included in clinical trials. On the other hand, among the best-ranked 100 predictions, eight were cytotoxic drugs.

## 6. Discussion

Through this review, it was our intention to provide an overview of studies relying on omics data and/or network analysis to improve the knowledge of the molecular mechanisms triggered by SARS-CoV-2 infection and underlying the organ and tissue damage. Globally, the semantic analysis proposed at the beginning of our work foreshadowed the poor representation of studies dedicated to this area of research. In fact, following the exploration of this landscape, we obtained few manuscripts specifically targeting the molecular profiles of tissues and organs. If, on the one hand, it was shown that these topics are poorly investigated, on the other hand, it highlights the need to fill this gap.

The omics analyses already carried out represent an important source of information for retrospective studies. Similarly, it is relevant to consider the possibility of investigating today tissue and organ samples collected from COVID-19 patients during the pandemic. This delay is probably due to several factors. Among them, social restrictions have affected every type of work activity, including that dedicated to scientific research. In addition, as highlighted by our semantic analysis, the priorities during the early stages of the pandemic were linked to the pressure suffered by hospitals, as well as by healthcare workers. In other words, at least in the beginning, strict precautionary measures and resources were put forward by most nations in order to mitigate transmission and decrease fatality rates. Thus, healthcare workers, including doctors, spent more time and effort in treating patients with already available tool than in collecting samples from deceased patients. The collection of these, moreover, was only possible in specialized centers. On the other hand, many researchers were forced to work at home, focusing more on scientific writing than on analytical activities [113].

Except for cell lines analyzed at different times after infection [41,42], the restrictions caused by the lockdown most probably impacted also the collection of longitudinal data. The potential associated with this type of study [114], and the combination of omics data and network strategies, is still to be exploited and could allow new insights into the physiological mechanisms dysregulated by SARS-CoV-2 infection and the pathological ones triggered by it. The data published by Nie et al. certainly represent the best example of a dataset, to our knowledge, that could be further used to reconstruct PPI network models [11]. Even if the characterized profiles concern only the proteomic analysis, the dataset covers several organs, providing, for each of them, a widely represented proteome. To our knowledge, the rest of the panorama offers omics profiles from biofluids [29,30,31,32,38,39] or cell lines [40,41,42,43], which, while not representing the complexity of a tissue or an organ, still play a significant role in accumulating knowledge and having as complete a picture as possible. Indeed, although the role of omics and network analyses does not significantly emerge from the titles of the published papers (Figure 2), the picture offered by our overview highlights the importance of these approaches in discovering new molecular relationships that intervene in the host–virus interaction, as well as in opening up new strategies that allow the faster development or reuse of pharmaceuticals.

The completeness of the omics profiles used to reconstruct the network models can certainly impact the consistency of the models themselves and the outcomes of their functional and topological analysis. As for COVID-19 investigation, these strategies have been much more widely adopted and exploited in combination with RNAseq data [23,25,26,74,75,76,77,78,79,80,81,82,83,84,85], and specifically in the context of network pharmacology studies. As a result of using traditional medicine to treat COVID-19 patients, the number of retrospective studies aimed at discovering bioactive compounds and potential gene targets for therapy has clearly emerged. The compelling need for pharmacological solutions has in fact stimulated different and complementary approaches, including drug repurposing, drug design platforms, in vitro assays and animal models, up to the development of vaccines [115]. Most of these studies were based on a similar pipeline relying on network topology. The identification of protein hubs has proven useful in prioritizing the candidate targets for drug treatments in which to invest time for further study and validation [23,24,25,26,74,75,76,77,78,79,80,81,82,83,84,85,86,89,90,91,92,93,94]. The saving of time certainly represents a vitally important element in emergency situations, such as the one experienced due to COVID-19.

Though using different strategies, other studies have relied on topology for drug reuse purposes. Like the betweenness, or other centralities for the selection of hubs, the proximity distance was found to be useful to rank the potential benefits of ready-to-use drugs [20,56,111]. The interest and relevance of this research area are underlined by the development of other algorithms both before [104,105] and after the pandemic [107,108]. Indeed, beyond COVID-19, which has been an extraordinary condition, the increase in drug development costs combined with a significant decline in the number of new drug approvals increases the need for innovative approaches to identifying targets and predicting drug efficacy.

All repurposing strategies, whether they are in silico or in combination with experimental data, suffer from some limitations that emerge for different reasons. In the case of experimental data, and in particular of clinical proteomics, the standardization of methods remains the main challenge in obtaining reproducible results [116]. The completeness of the characterized profiles, combined with that of accurate protein–protein interaction network models [117,118], is a further factor that certainly can influence the results obtained. In this scenario, in vitro drug screening is the first step in validating potential drugs selected by in silico approaches. However, conflicting evidence may be also the result of the investigation of different cell lines, different readout times, different drug concentrations or different viral MOI. As reported by Kuleshow [119], different laboratories across the world, different assays or different models tested produce different lists of drugs and targets with a small overlap (Figure 6). As shown in this figure, a larger number of drugs was shared in studies performed by the same authors, i.e., Pickard et al. [120] and Xiao et al. [121]. Nevertheless, some drugs, including remdesivir, chloroquine, hydroxychloroquine and mefloquine, have been highlighted in multiple studies. Remdesivir is a good example of a direct-acting antiviral drug that inhibits viral RNA polymerase. Hence, it was found through drug-repurposing methods relying on docking patterns. In contrast, pipelines based on network models could also identify drugs that target host proteins, like dexametasone [20], which is commonly used in hospitalized patients [122]. Finally, it is also important to highlight that some drugs tested in cells like VeroE6 could have different efficacy in human cells, like Caco-2 or others. On the other hand, top-ranking drugs, like ritonavir, do not show effects by in vitro screening but dozens of clinical trials are exploring its potential efficacy in patients. Similarly, drugs effective in vitro may not replicate in vivo, as observed for chloroquine and hydroxychloroquine, which work only in combination with azithromycin [123].

## 7. Conclusions

The picture offered by our review work is that the investigation area of omics and systems biology has been little explored, with the exception of studies dedicated to traditional medicine and network pharmacology. This trend is probably in accordance with the ongoing health emergency, and the prioritization of effective drugs even before understanding the mechanisms underlying the infection or triggered by it. However, this has still allowed us to devote our efforts to the development and application of innovative and inexpensive methods to evaluate the reuse of ready-to-use medicines. The goal of drug repurposing is to prioritize all available drugs, allowing us to limit experimental efforts only to the top-ranking compounds—hence improving efficiency and resource utilization. As noted in our discussion, we are aware of the limitations that can arise from these approaches, which can be further exacerbated by the use of different cellular models or those derived from organisms other than humans. We also know that clinical trials, aimed at discovering or verifying the effects of a new drug or an existing one tested for new methods of therapeutic use, are divided into various phases and require timescales that can even exceed several years. Furthermore, each country has different timescales for the drug to be marketed. Nevertheless, the health emergency caused by SARS-CoV-2 has led us to reduce the approval times for clinical use, and the development of vaccines is the most striking example. For this purpose, the combination of omics technologies and network approaches remains a promising strategy, despite all the aspects that need to be improved. They therefore represent a tool to be further used in discovering new treatments for various pathologies. This is a non-trivial opportunity that deserves to be discussed and explored, not only in the event of an emergency but also for those rare diseases that suffer from low investments regarding the development of dedicated drugs. This also applies to the knowledge that has emerged from studies based on traditional medicine. In fact, the discovery of new, potentially bioactive molecules should favor their investigation in order to speed up their clinical investigation and approval.

## Figures and Tables

**Figure 1 biology-12-01196-f001:**
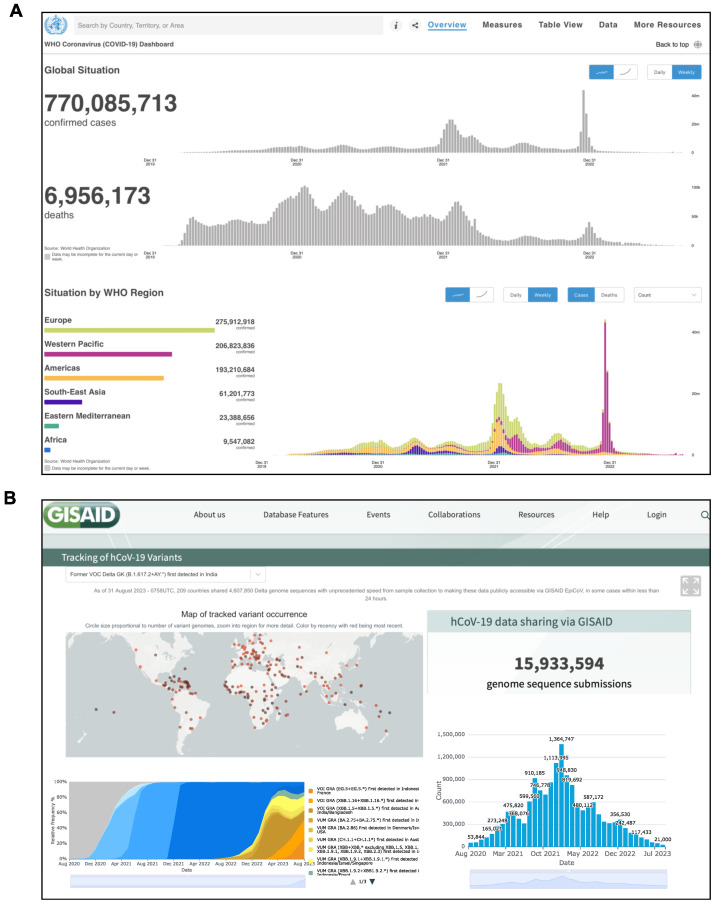
Snapshot of the official websites reporting SARS-CoV-2 data in real time. (**A**) World Health Organization (WHO) Coronavirus (COVID-19) Dashboard (https://COVID-19.who.int/ (accessed on 1 August 2023)). (**B**) GISAID Repository (https://gisaid.org/hcov19-variants/ (accessed on 1 August 2023)).The GISAID Initiative promotes the rapid sharing of data from all influenza viruses and the coronavirus causing COVID-19. This includes genetic sequences and related clinical and epidemiological data associated with human viruses, and geographical as well as species-specific data associated with avian and other animal viruses, to help researchers to understand how viruses evolve and spread during epidemics and pandemics.

**Figure 2 biology-12-01196-f002:**
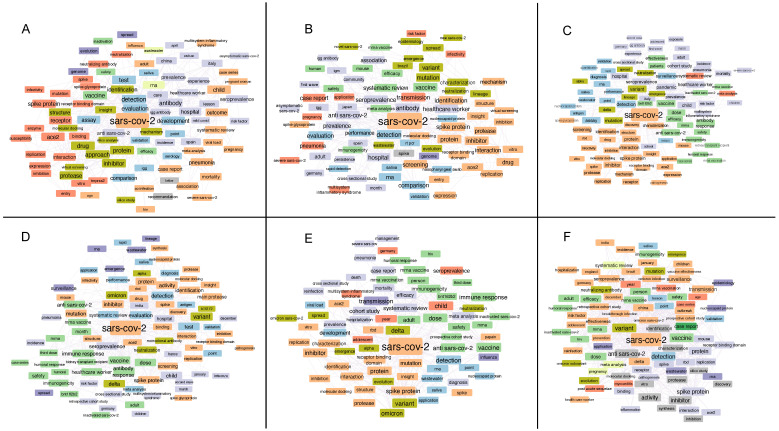
Term associations from title of manuscripts published in (**A**) 2nd semester 2020, (**B**) 1st semester 2021, (**C**) 2nd semester 2021, (**D**) 1st semester 2022, (**E**) 2nd semester 2022 and (**F**) 1st semester 2023. All manuscripts were retrieved in PubMed (https://pubmed.ncbi.nlm.nih.gov (accessed on 1 May 2023)) by searching for “SARS-CoV-2” in the “Title” field. Title terms were associated using the VOSviewer software (www.vosviewer.com) (accessed on 1 August 2023) and the 100 top-ranked associations were displayed using the Cytoscape platform. Different colors show clusters of terms most correlated.

**Figure 3 biology-12-01196-f003:**
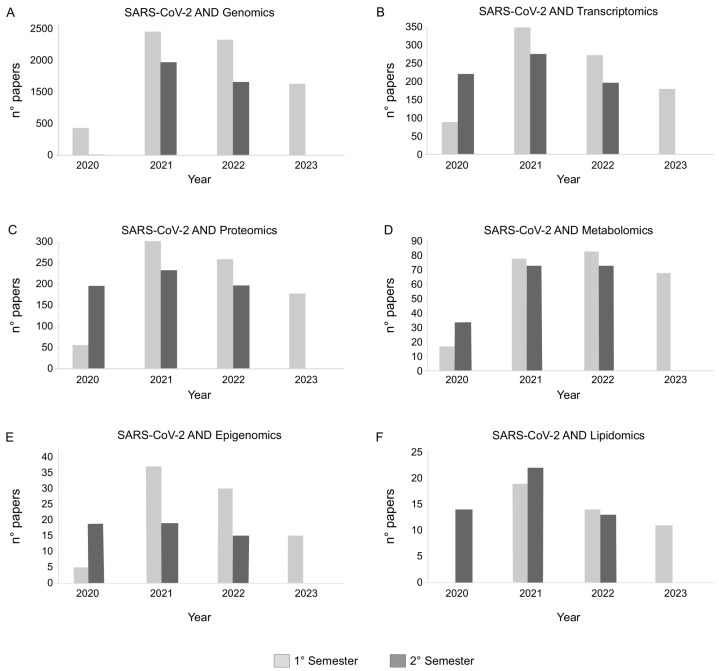
Number of papers published per semester from 2020 to 2023 and found in PubMed (https://pubmed.ncbi.nlm.nih.gov (accessed on 1 May 2023)) by searching for SARS-CoV-2 AND (**A**) Genomics, (**B**) Transcriptomics, (**C**) Proteomics, (**D**) Metabolomics, (**E**) Epigenomics or (**F**) Lipidomics.

**Figure 4 biology-12-01196-f004:**
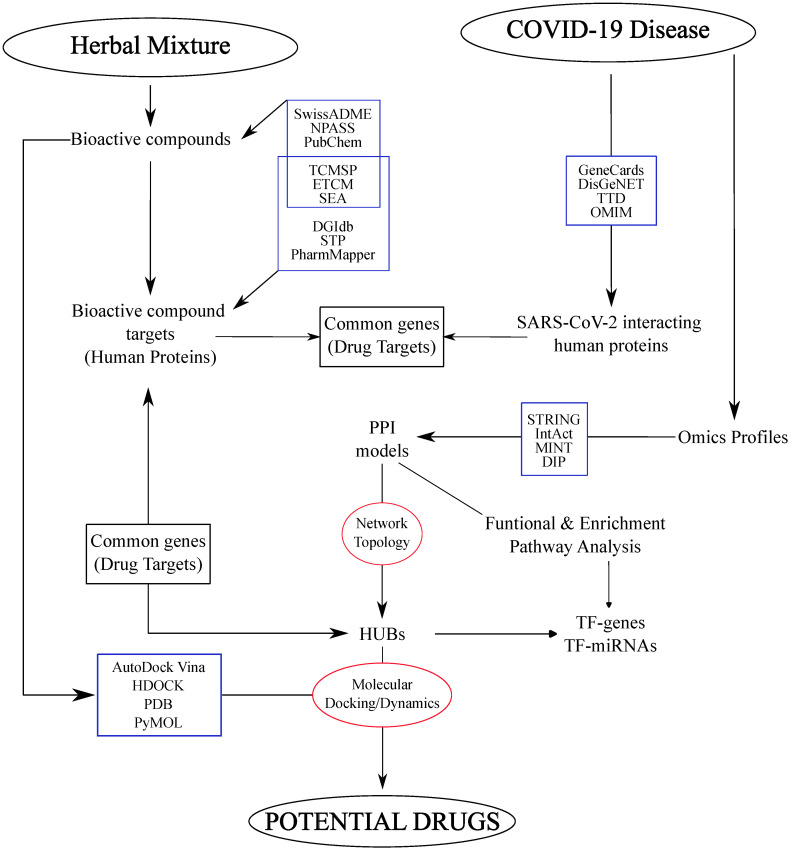
Workflow summarizing the main steps in discovering gene targets and herbal bioactive compounds as potential drugs to treat COVID-19 patients. Starting with herbal mixtures, the authors retrieved the bioactive compounds present there and the corresponding genes that they target. These genes were matched with those interacting with SARS-CoV-2 proteins to select compound/gene combinations of interest. The further identification of potential drug targets (HUBs) was performed by the topological analysis of network models reconstructed from the omics profiles characterized by COVID-19 patient samples. Finally, the selected combinations compound/gene(HUB) were in silico validated by molecular docking/dynamics. The blue rectangles indicate the main databases used, while the red circles indicate the in silico approaches.

**Figure 5 biology-12-01196-f005:**
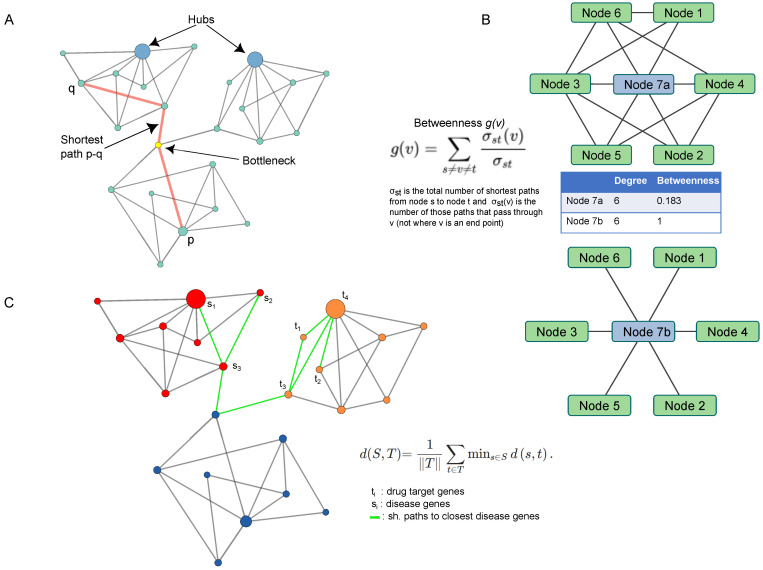
Network topology and parameters used to select hubs and candidate drug targets. (**A**) Scheme representing hubs, bottlenecks and shortest path. (**B**) Betweenness centrality and its variation in network models taken as an example. (**C**) Proximity distance. Red and orange nodes indicate genes belonging to disease and drug target modules, respectively. In green, the shortest paths between disease genes (S1, S2 and S3) and the drug target genes (t1, t2, t4 and t4) are shown. Node size is proportional to node degree.

**Figure 6 biology-12-01196-f006:**
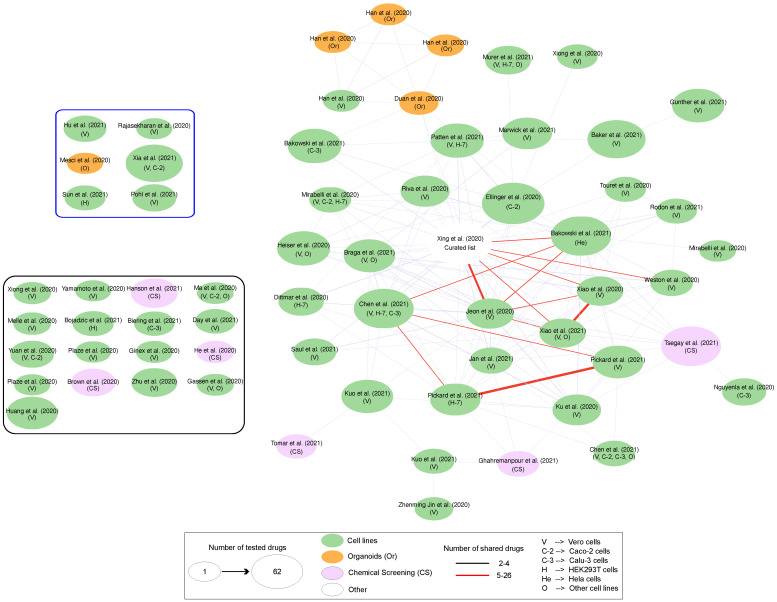
Overlapping of drugs selected in different studies reported in https://maayanlab.cloud/covid19/ (accessed on 1 August 2023) [119]. Datasets (circles) are connected if they share at least 2 drugs. Blue rectangles indicate datasets that do not share any drugs, while black ones indicate datasets that share only one drug. Circle size is proportional to the number of drugs found. Red edges indicate studies sharing at least 5 drugs.

**Table 1 biology-12-01196-t001:** Collection of web resources and databases useful for network pharmacology; OSP: Open-Source Project.

Database/Link	Description	License	Ref.
DGIdb dgidb.org (accessed on 1 August 2023)	Web resource that provides information on drug–gene interactions and druggable genes from publications, databases and other web-based sources	OSP	[59]
TTD db.idrblab.net/ttd/ (accessed on 1 August 2023)	Database that provides information about the known and explored therapeutic protein and nucleic acid targets, the targeted disease, pathway information and the corresponding drug target	Free	[60]
NPASS bidd.group/NPASS/index.php (accessed on 1 August 2023)	It integrates species sources of natural products and connects natural products to biological targets via experimentally derived quantitative activity data	Free	[61]
SwissADME swissadme.ch (accessed on 1 August 2023)	It allows predictive models for physicochemical properties, pharmacokinetics, ADME parameters, druglikeness and medicinal chemistry friendliness of one or multiple small molecules to support drug discovery	Free	[62]
STP swisstargetprediction.ch (accessed on 1 August 2023)	It estimates the most probable macromolecular targets of a small molecule, assumed as bioactive from three different species (*Homo sapiens*, *Mus musculus*, *Rattus norvegicus*)	Free	[63]
SEA sea.bkslab.org (accessed on 1 August 2023)	It quantitatively groups and relates proteins based on the chemical similarity of their ligands. It can be used to rapidly search large compound databases and to build cross-target similarity maps	Free	[64]
PharmMapper lilab-ecust.cn/pharmmapper/ (accessed on 1 August 2023)	It is a web server that allows the identification of potential small molecule targets using a pharmacophore mapping approach. The server hosts a large repertoire of pharmacophore models annotated from various sources and it finds the best mapping poses of the query molecule against all the pharmacophore models in the database	Free	[65]
TCMSP cmsp-e.com/tcmsp.php (accessed on 1 August 2023)	It captures the relationships between drugs, targets and diseases as well as pharmacokinetic properties for natural compounds	Free	[66]
PubChem pubchem.ncbi.nlm.nih.gov (accessed on 1 August 2023)	It is an open chemistry database that mainly contains small molecules, but also nucleotides, carbohydrates, lipids, peptides and chemically modified macromolecules. It collects information on chemical structures, identifiers, chemical and physical properties, biological activity, patents, health and safety, toxicity data and many others	OSP	[67]
GeneCards genecards.org (accessed on 1 August 2023)	It is a searchable integrative database that provides comprehensive user-friendly information on all annotated and predicted human genes. It automatically integrates gene-centric data from approximately 150 web sources, including genomic, transcriptomic, proteomic, genetic, clinical and functional information	OSP	[68]
OMIM omim.org (accessed on 1 August 2023)	It is a comprehensive, authoritative compendium of human genes and genetic phenotypes that is freely available and updated daily. The full-text, referenced overviews in OMIM contain information on all known Mendelian disorders and over 16,000 genes. OMIM focuses on the relationships between phenotypes and genotypes	OSP	[69]
ETCM tcmip.cn/ETCM/ (accessed on 1 August 2023)	It includes the most commonly used herbs and formulas of Traditional Chinese Medicine, as well as their ingredients, to explore the relationships or build networks among TCM herbs, formulas, ingredients, gene targets and related pathways or diseases	Free	[70]
STRING string-db.org (accessed on 1 August 2023)	It aims to integrate all known and predicted associations between proteins, including both physical interactions and functional associations. To achieve this, STRING collects and scores evidence from a number of sources: (i) automated text mining of the scientific literature, (ii) databases of interaction experiments and annotated complexes/pathways, (iii) computational interaction predictions from co-expression and from conserved genomic context and (iv) systematic transfers of interaction evidence from one organism to another. The upcoming version 11.5 of the resource will contain more than 14,000 organisms	Free	[71]
STITCH 2 stitch.embl.de (accessed on 1 August 2023)	It aims to integrate the data dispersed throughout the literature and various databases of biological pathways, drug–target relationships and binding affinities. In STITCH 2, the number of relevant interactions is increased by the incorporation of BindingDB, PharmGKB and the Comparative Toxicogenomics Database. The resulting network can be explored interactively or used as the basis for large-scale analyses. STITCH 2 connects proteins from 630 organisms to over 74,000 different chemicals, including 2200 drugs	Free	[72]
CHEMBL https://www.ebi.ac.uk/chembl/ (accessed on 1 August 2023)	ChEMBL is a manually curated database of bioactive molecules with druglike properties. It brings together chemical, bioactivity and genomic data to aid the translation of genomic information into effective new drugs	Free	[73]
Pathguide pathguide.org (accessed on 1 August 2023)	It contains information about 702 biological pathway-related resources and molecular interaction-related resources. Databases that are free and those supporting BioPAX, CellML, PSI-MI or SBML standards are indicated	Free	

## Data Availability

Not applicable.

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
