# Peer review of "Integration of Omics Data and Network Models to Unveil Negative Aspects of SARS-CoV-2, from Pathogenic Mechanisms to Drug Repurposing"

_biology, 2023, doi:10.3390/biology12091196_

Round 1

Reviewer 1 Report

The authors present a high-level review of how multiomics and network analyses can be used to dive deeply into large volumes of metadata, with the purpose of identifying drug targets or repurposed drugs for the treatment of COVID-19.  The Introduction contains a good summary of COVID-19 prevalence, entry, and correlation to human disease states.  This section nicely sets up the framework for the subsequent review.  This manuscript is a well-summarized review of proteomics and metabolomics results that were obtained from human samples, which shows multiple pathways differentially affected by SARS-CoV-2 in various compared populations.  The authors also provide an informative basic explanation of network topology (hubs, bottlenecks, and betweenness) that is understandable to the lay reader.  A few revisions are recommended, mostly related to wanting more application and translation of the materials reviewed to treatment outcomes of people with COVID-19.

Lines 150-174 review differential gene regulation in cell culture between infected and uninfected cells.  Can these findings be related to human tissues and fluid multiomics analyses to demonstrate the potential translative ability of cell culture results when evaluating drugs?

What prevented progress of omics-identified repurposed drugs to be approved as treatments (or show positive clinical results)?  For example, carvedilol was identified as a potential therapy and the authors noted it as effective for treating COVID-19 patients, but it is not an approved drug.  What is the missing gap in translation of efficacy noted to clinical approval?

The drugs noted as having the highest proximity scores were ultimately shown as ineffective as treatments or not authorized for COVID-19 (i.e chloroquine, methotrexate, hydroxychloroquine, omeprazole).  Can the authors demonstrate how proximity measure yielded drugs that were ultimately approved therapies for SARS-CoV-2?  For example, does the model yield results showing precursors to nirmatrelvir (such as other protease inhibitors from other antiviral treatments – such as boceprevir), remdesivir, or as anti-inflammatory drugs commonly used in hospitalized patients (i.e. dexamethasone).

The semantic and titles analyses in the Discussion are not presented earlier in the main text, nor are they well-related to the main text when presented in the Discussion.  Please tie these analyses to the main body of the review more thoroughly or introduce these concepts in the main body – rather than introduce a new topic in the Discussion.

Last sentence (lines 445-448) is unclear and written with phrases rather than a coordinated sentence.  Please rephrase to clarify the conclusion intended.

The overall quality of the manuscript will benefit from further review and revision by the editorial staff, as there are minor but frequent changes in grammar needed.  Some examples include COVID19 to COVID-19; hearth to heart; death vs dead, contribute to contribution, no need to write out the abbreviation for SARS-CoV-2 late in the paper (lines 361-362), as well as sentence structure and proper use of prepositions.

Author Response

Q1: Lines 150-174 review differential gene regulation in cell culture between infected and uninfected cells.  Can these findings be related to human tissues and fluid multiomics analyses to demonstrate the potential translative ability of cell culture results when evaluating drugs?

A1: We thank R1 for this question that allowed us to improve our review with a comment on multiomics analyses of tissues, biofluids and cell lines (see L202-210). We believe that results from cell lines can be related to human tissues and biofluids, thus they can be used for evaluating both disease states and drug efficacy. However, it is important to underline the limitation of analyzing a single cell line, decontextualized from its natural environment made up of molecular and cellular interactions. The analysis of biofluids, mainly serum and plasma, gives results which reflects the state of different sources, including cell lines and tissues. Thus, they provide a more systemic view. In this scenario, specific tissues certainly represent well defined complex systems whose multiomics profiles can be better associated with a disease state, as well as to therapeutic efficacy. Moreover, a further comment was added at the end of our Discussion (see L491-493) where we highlighted the importance of specifying the cell type origin used for screening drugs, like Vero or Caco.

Q2: What prevented progress of omics-identified repurposed drugs to be approved as treatments (or show positive clinical results)?  For example, carvedilol was identified as a potential therapy and the authors noted it as effective for treating COVID-19 patients, but it is not an approved drug.  What is the missing gap in translation of efficacy noted to clinical approval?

A2: We thank R1 for this excellent observation that allowed us to comment the gap between the drug efficacy emerged by in silico studies and its translation to clinical practices. The reasons of this gap are for sure different. As we added to our Discussion “We are well aware that clinical trials, aimed at discovering or verifying the effects of a new drug or an existing one tested for new methods of therapeutic use, are divided into various phases and require timescales that can even take several years. Furthermore, each country has different timescales for the drug to be marketed. Despite this, the health emergency caused by COVID19 has exceptionally led us to reduce the approval times for clinical use, and the development of vaccines is the most striking example” (see L522-528).

Concerning carvedilol, it was approved for medical use in the USA in 1995. In particular, it was initially designed for managing hypertension and coronary artery disease. The study we cited concerning its potential effectiveness against COVID19 was recently published (October 2022). The authors there evidenced a significant network proximity to SARS-CoV-2 host factors for 23 drugs, including carvedilol that showed clinical benefits for COVID-19 patients. In fact, using two large independent COVID-19 patient databases, they found that usage of carvedilol, was associated with a lowered risk (17–20%) of a positive COVID-19 test, validating the trustworthiness of the proximity measure (see L269-271). Moreover, following the pandemic emergency and the results obtained by these studies, we understood that these computational approaches will have to be exploited to improve knowledge of drugs already available and therefore speed up their use in clinical practice should it become necessary to face another crisis. This also applies to the knowledge that emerged from studies based on traditional medicine. In fact, the discovery of new potentially bioactive molecules should favor their investigation in order to speed up their clinical approval.

Q3: The drugs noted as having the highest proximity scores were ultimately shown as ineffective as treatments or not authorized for COVID-19 (i.e chloroquine, methotrexate, hydroxychloroquine, omeprazole).  Can the authors demonstrate how proximity measure yielded drugs that were ultimately approved therapies for SARS-CoV-2?  For example, does the model yield results showing precursors to nirmatrelvir (such as other protease inhibitors from other antiviral treatments – such as boceprevir), remdesivir, or as anti-inflammatory drugs commonly used in hospitalized patients (i.e. dexamethasone).

A3: We thanks R1 for this question. Following that, we integrated our Discussion with some comments about remdesivir and dexamethasone (L498-502).

Q4: The semantic and titles analyses in the Discussion are not presented earlier in the main text, nor are they well-related to the main text when presented in the Discussion.  Please tie these analyses to the main body of the review more thoroughly or introduce these concepts in the main body – rather than introduce a new topic in the Discussion.

A4: We have followed R1's lead, and a new paragraph specifically referring to semantic analysis has been improved and moved to the top of our review (L75-100).

Q5: Last sentence (lines 445-448) is unclear and written with phrases rather than a coordinated sentence.  Please rephrase to clarify the conclusion intended.

A5: We thank R1 for this suggestion. We rephrased the sentence (L531-533).

Q6: The overall quality of the manuscript will benefit from further review and revision by the editorial staff, as there are minor but frequent changes in grammar needed.  Some examples include COVID19 to COVID-19; hearth to heart; death vs dead, contribute to contribution, no need to write out the abbreviation for SARS-CoV-2 late in the paper (lines 361-362), as well as sentence structure and proper use of prepositions.

A6: We thank R1 for these suggestions. We proceeded to make the corrections and revise the English.

Reviewer 2 Report

The authors assembled a nice review on the topic of integration omics data to network models, taking advantage from SARS CoV2 plenty of data and models. They also claim coverage from viral-host pathogenic mechanisms to drug repurposing efforts. The review is well written and can be read with easiness. Reference are not thorough but 111 references are good starting points.

In my opinion there are few points that authors should considering adding for publication, as it is now the review is not complete.

Often reviews cannot be really "complete", but the lack of following points can make the difference between a good review from an anonymous one.

1) Stay focused. Even if cited as example for good performance a cardio-centric study should only be cited "en-passant" and as it is now is not really fitting the scope.

There are tons of "successful" disease-drug interactions network out there which still need experimental validations.

2) On the other side, repurposing has been touched "en-passant" by the review as it missed the opportunity to dig deep in the real problem of repurposing when used for SARS CoV-2. A brief look into https://maayanlab.cloud/covid19/ could illustrate it: select any of the 67 datasets collected by the Mayyan's Team and generate a Venn diagram out of it. Hitlists are only marginally overlapping, indicating that minor changes in the assays will deliver different molecules as hits, deeply affecting any network generated by those assays (the review cited only Riva et al.. Can omics integration be effective there? On this problem, review did not spend a word. To me a crucial lack of prospective for preclinical SARS CoV-2 studies.

3) Thirdly, authors did not spend any sentence on limitations of these methods to warn readers on exceeding expectations. No words on the evident lack of positive predictions for the top ten drugs predicted as good in the cited work, where still Chloroquine and Hydroxychloroquine were present. 

4) Fourth, the generation of networks based on semantic extraction of interactions from literature papers lacks of a discussion for counterfactual evidences. Often two different conflicting evidences reported are just the results of different cell lines, different readout times, different drug concentrations, different viral MOI. How did the network authors went about to solve literature conflicting evidences ? Could we have a table (even if incomplete) of the networks that paper took into account? A summary table with the actual evidences, for instance, exposed at pag 5 (lines 132-140) could be of higher value than Figure 3 which, in turn, does not seem to be tightly linked to text.

Notwithstanding this critical points, review has strong points on longitudinal side. Fig 5 is a nice evidence of the historical changes in literature during the years analyzed. Unfortunately, authors did not dig too much about lack of longitudinal omics data (both on clinical and preclinical side).

All in all, I don't think that the review in present status is enough to be published on IF=7 journal, but it would need some extra work in the directions above given.

Minor enhancements:

a) Figure 1 would need a better resolution

b) Add CHEMBL to table 1. It has been a noticeable source of annotations for almost all drug network so far published

Best regards

Author Response

Q1: Stay focused. Even if cited as example for good performance a cardio-centric study should only be cited "en-passant" and as it is now is not really fitting the scope. There are tons of "successful" disease-drug interactions network out there which still need experimental validations.

A1: We thank R2 for this comment and following it we reduced that part. (L387-394)

Q2: On the other side, repurposing has been touched "en-passant" by the review as it missed the opportunity to dig deep in the real problem of repurposing when used for SARS CoV-2. A brief look into https://maayanlab.cloud/covid19/ could illustrate it: select any of the 67 datasets collected by the Mayyan's Team and generate a Venn diagram out of it. Hitlists are only marginally overlapping, indicating that minor changes in the assays will deliver different molecules as hits, deeply affecting any network generated by those assays (the review cited only Riva et al.. Can omics integration be effective there? On this problem, review did not spend a word. To me a crucial lack of prospective for preclinical SARS CoV-2 studies.

A2: We thanks R2 for this interesting suggestion. Take this source as reference we integrated our Discussion with the limitations concerning the drug repurposing and the poor reproducibility generated by different assays and models tested (L485-506). In addition, we have further extended the description of some studies focused on drug repurposing (L394-421). In addition, a new Figure6 summarizing the manuscripts reported in https://maayanlab.cloud/covid19/ has been added.

Q3: Thirdly, authors did not spend any sentence on limitations of these methods to warn readers on exceeding expectations. No words on the evident lack of positive predictions for the top ten drugs predicted as good in the cited work, where still Chloroquine and Hydroxychloroquine were present. 

A3: As reported for the question raised in point 2 by the R2 we integrated our Discussion with limitations concerning the drug repurposing (L485-506).

Q4: Fourth, the generation of networks based on semantic extraction of interactions from literature papers lacks of a discussion for counterfactual evidences. Often two different conflicting evidences reported are just the results of different cell lines, different readout times, different drug concentrations, different viral MOI. How did the network authors went about to solve literature conflicting evidences ? Could we have a table (even if incomplete) of the networks that paper took into account? A summary table with the actual evidences, for instance, exposed at pag 5 (lines 132-140) could be of higher value than Figure 3 which, in turn, does not seem to be tightly linked to text.

A4: We thanks the R2 for these comments. The goal of our semantic analysis was simply to find the association between words starting from manuscript titles containing SARS-CoV-2 as term. Thus, by itself, this can be a limitation. We agree conflicting evidences are just the results of different cell lines, different readout times, different drug concentrations, different viral MOI. However, in our opinion in can’t emerge by the semantic analysis we performed. As suggested by R2 at point 2 we extended our Discussion with the limitation of drug repurposing and its validation (L485-506).

As for figure 3, now Figure 4, it aims to describe a scheme/workflow used in several studies based on traditional medicine. As reported in our overview, the network reconstructed in these studied were data-derived. Thus, they change study by study. However, as indicated by figure 3 (now 4) the main source for these models was STRING. Finally, to improve the value of the figure 3 (now 4) we integrated its legend with a deeper description and we added a further reference in the text (see new Figure4).

Q5: Notwithstanding these critical points, review has strong points on longitudinal side. Fig 5 is a nice evidence of the historical changes in literature during the years analyzed. Unfortunately, authors did not dig too much about lack of longitudinal omics data (both on clinical and preclinical side).

A5: We thank the R2 for this positive comments. Concerning the lack of longitudinal omics data, we integrated our Discussion with possible explanations (L443-445).

Q6: Figure 1 would need a better resolution

A6: We thank R2 for this indication. We are confident to solve that with the editorial staff.

Q7: Add CHEMBL to table 1. It has been a noticeable source of annotations for almost all drug network so far published

A7: We thank R2 for this suggestion. We added CHEMBL to Table1.

Round 2

Reviewer 2 Report

Authors have solved some problems